# Stop the Pop: A Mixed-Methods Study Examining Children’s Physical and Emotional Responses during Three Days of Sugary Drink Cessation

**DOI:** 10.3390/nu14071328

**Published:** 2022-03-22

**Authors:** Jasmine H. Kaidbey, Kacey Ferguson, Sabrina E. Halberg, Caroline Racke, Amanda J. Visek, Ashley N. Gearhardt, Laura M. Juliano, William H. Dietz, Jennifer Sacheck, Allison C. Sylvetsky

**Affiliations:** 1Department of Exercise and Nutrition Sciences, Milken Institute School of Public Health, The George Washington University, Washington, DC 20052, USA; kaidbey@gwmail.gwu.edu (J.H.K.); kacey_ferguson@email.gwu.edu (K.F.); halbergs@gwmail.gwu.edu (S.E.H.); caroline_racke@gwmail.gwu.edu (C.R.); avisek@gwu.edu (A.J.V.); jsacheck25@email.gwu.edu (J.S.); 2Department of Psychology, University of Michigan, Ann Arbor, MI 48109, USA; agearhar@umich.edu; 3Department of Psychology, American University, Washington, DC 20016, USA; juliano@american.edu; 4Sumner M. Redstone Global Center for Prevention and Wellness, Milken Institute School of Public Health, The George Washington University, Washington, DC 20052, USA; bdietz@email.gwu.edu

**Keywords:** sugar-sweetened beverages, soda, soft drinks, juice drinks, beverage consumption, behavior, youth, adolescents, interventions, home environment

## Abstract

Despite public health efforts to reduce sugary drink consumption, children’s intake continues to exceed recommendations. While numerous barriers to lowering sugary drink consumption have been identified, aversive feelings during sugary drink cessation may further challenge sustained reduction in children’s sugary drink consumption. Herein, we describe “Stop the Pop”, an intervention to examine children’s physical and emotional responses during three days of sugary drink cessation. Children (*n* = 150) ages 8–14, who reported habitual consumption of ≥12 ounces of sugary drinks daily, were instructed to avoid sweetened beverages for three days. At baseline and on each day of cessation, children completed a daily feelings questionnaire, and a subset of children (*n* = 30) also completed a qualitative interview following cessation. During sugary drink cessation, children reported physical and emotional improvements, including being less tired, angry, and annoyed; having less trouble sleeping; and less frequently arguing with others, getting in trouble, and getting mad. However, unfavorable responses, such as mood disturbances and having less energy, were reported by some participants. Our results suggest that children who habitually consume sugary drinks may experience physical and emotional improvements during short-term sugary drink cessation, although longer-term examination is needed and inter-individual variability in responses to cessation warrants further study.

## 1. Introduction

Sugary drinks are the main source of added sugar in the American diet [1], and about three-quarters of children and adolescents aged 2–19 in the United States (US) report consuming sugary drinks on a given day [2]. Sugary drink consumption is associated with the development of obesity and cardiometabolic disease [3,4], as well as a constellation of other unfavorable behaviors, including inadequate water intake, insufficient sleep, and high consumption of energy-dense, nutrient-poor foods [1].

Given the well-established adverse health effects of excess sugary drink consumption [5], reducing sugary drink intake among children is paramount. However, children’s sugary drink consumption continues to exceed public health recommendations [2,6], in part due to a variety of biological, psychosocial, and environmental factors that promote excess sugary drink consumption [7,8]. While the extent to which sugar is “addictive” in humans remains controversial [9], sugar activates homeostatic circuits that regulate energy balance, as well as hedonic circuits that make food pleasurable, some of which overlap with neural reward circuits activated by addictive drugs [10]. In our recent qualitative study investigating multi-factorial reasons for sugary drink consumption among children, participants described patterns of sugary drink use that were, in some cases, consistent with substance-use disorder symptomology, including consumption of sugary drinks to satisfy cravings and reduce negative affect, devoting significant time and effort to obtain sugary drinks, and continued use of sugary drinks despite knowledge of their negative health effects [11]. In a separate study of children’s sugary drink consumption behaviors using concept mapping [7], participants reported a variety of reasons for sugary drink use, including improved physical (e.g., less sleepiness and enhanced performance), cognitive (e.g., better attention), and emotional (e.g., greater happiness) states. We also previously conducted in-depth qualitative interviews with parents about their child’s sugary drink intake behaviors, during which parents described that their children developed aversive physical and/or emotional responses (e.g., anger, aggression, headaches, and mood disturbance) when sugary drinks were restricted [12].

While these findings suggest that aversive physical and emotional responses to sugary drink restriction may pose a barrier to sugary drink cessation among children, only one exploratory study to date has experimentally examined responses to sugary drink cessation among youth [13]. Although that study provided compelling preliminary evidence that youth may experience aversive physical and emotional responses to sugary drink cessation, the sample size was small, and only adolescents with overweight or obesity who reported high sugary drink consumption (≥3 servings per day) were enrolled. We aimed to examine physical and emotional responses to three days of sugary drink cessation in a larger sample of racially and ethnically diverse children, and we hypothesized that children would report aversive physical and emotional responses to sugary drink cessation [11]. Herein, we describe the design and findings of “Stop the Pop”, an entirely virtual single-arm intervention study designed to examine children’s physical and emotional responses during three days of sugary drink cessation.

## 2. Materials and Methods

The study protocol (NCR202124) was reviewed and approved by the Institutional Review Board at The George Washington University. Informed consent was obtained from the child’s parent/guardian, and all children provided assent prior to beginning the study procedures.

### 2.1. Participant Recruitment and Eligibility

Children aged 8–14 were recruited from across the continental US. Recruitment took place from November 2020 through June 2021 and was carried out entirely virtually, using social media, school, and community organization listservs, and parent-targeted study advertisements created by a professional recruitment agency. Interested parents completed a brief electronic eligibility survey administered via Qualtrics™, and eligible volunteers were subsequently contacted by the research team to schedule a virtual study visit. Inclusion criteria were the parent or guardian (hereafter parent) reporting that their child (1) was between 8 and 14 years of age and (2) consumed ≥12 ounces of sugary drinks (e.g., regular (non-diet) sodas, juices, sweet teas, and sports drinks) per day. Children taking medication(s) for allergies and/or asthma, and/or had a history of migraines, were excluded to reduce confounding with assessment of responses to sugary drink cessation. Based on power analyses using differences in overall “symptoms of withdrawal” reported by Falbe et al. before and after three days of sugary drink cessation [13], the target sample size was 150 participants.

### 2.2. Study Procedures

Eligible participants were scheduled for a virtual study visit (using Zoom™) with a trained research assistant (RA). After providing parental consent and child assent electronically, the parents were asked to complete a demographic questionnaire on behalf of their child.

#### 2.2.1. Measures

Children were asked to report their habitual beverage consumption (during “a typical week”), using an adapted version of the beverage intake questionnaire (BEVQ-15) and with assistance from their parent, as needed. The BEVQ-15 is a validated instrument to assess habitual beverage intake in adults [14] and children [15] (significantly correlated with beverages reported in four-day food records in adults, *r* = 0.61, and 24-h recalls in children, *r* = 0.74), and its use is recommended for examining associations between beverage consumption and health-related outcomes. The adapted questionnaire queried the following beverage categories: water, 100% fruit juice, juice drinks, milk, soda, tea, coffee, energy drinks, and sports drinks, with separate survey items for caffeinated vs. caffeine-free and regular vs. diet subcategories, as appropriate. Participants were also instructed to report consumption of any other beverages not listed on the questionnaire. Images of beverage containers (e.g., 8-ounce cup, 12-ounce soda can, and 20-ounce vending machine bottle) were provided to facilitate participants’ accurate estimation of portion-sizes.

Following completion of the beverage questionnaire, children completed a daily feelings questionnaire to assess their physical feelings and emotions prior to beginning sugary drink cessation. The 37-item measure combined items from the Caffeine Withdrawal Symptom Questionnaire (CWSQ) and Processed Food Withdrawal Scale for Children (ProWS-C), both of which have been validated, along with additional items derived from our prior qualitative findings [12,16]. The CWSQ is a validated instrument to assess caffeine withdrawal symptoms in adults (Cronbach’s alpha = 0.89) [17], and the ProWS-C is a validated instrument for assessing parent-reported withdrawal symptoms from processed food avoidance among children (Cronbach’s alpha = 0.94) [18]. For each item, participants were asked to rate the extent to which they experienced the feeling over the past 24 h, using a 5-point Likert-style scale with the following response options: “Never”, “A little”, “Sometimes”, “Often”, and “A lot”.

### 2.2.2. “Stop the Pop” Intervention

After completion of the questionnaires, participants were counseled by an RA to (1) avoid all sweetened beverages (regardless of whether sugar-sweetened or artificially sweetened), including sodas, fruit drinks and juices, sports drinks, and sweetened milks, for three days (72 h) and (2) to instead consume only water, unsweetened seltzer, and/or unsweetened milk. Based on the participant’s reported habitual beverage intake, the RA provided personalized counseling for each child. For example, the RA explained that the child should not have “Sprite™, chocolate milk, or Gatorade™” if the child indicated that they habitually consumed these specific beverages. The three-day time frame for the present study was selected for two reasons: (1) acute withdrawal symptoms resulting from discontinuation of other substances, such as caffeine, occur in the first 48–72 h and peak 20–51 h following discontinuation [19]; and (2) our study was informed by the only other study on responses to sugary drink cessation in children conducted to date, as it used a similar design and examined sugary drink cessation over a three-day period [13]. Participants were also provided with a list of reminders via e-mail to encourage adherence to the intervention. During the subsequent three days of sugary drink cessation, participants were sent the daily feelings questionnaire (text or e-mail, per their preference) at the same time each day. Participants were instructed to complete the questionnaire as soon as possible upon receipt and to indicate (yes/no) whether they had consumed any sugary drinks in the last 24 h to assess intervention adherence. If the questionnaire was not completed within an hour of receipt, the participant’s parent was contacted by the research team with a reminder to complete the questionnaire. Participants received $50 as compensation for study participation following completion of their final daily feelings questionnaire.

### 2.2.3. Qualitative Interview

An in-depth qualitative interview about the participants’ experience in the study was conducted (also remotely via Zoom™) with each participant and their parent after the sugary drink cessation period, until saturation of the data was reached. Saturation was defined as “no new data, no new themes, no new coding”, as is consistent with the definition agreed upon by many qualitative researchers [20]. In instances where participants were siblings (there were five instances), multiple children were interviewed together to reduce parent burden. Qualitative interviews were conducted by one of three (A.C.S., J.H.K. and S.E.H.) trained interviewers, using a semi-structured guide, which included questions about the child’s experiences in the study and any physical, emotional, or behavioral changes observed during the three days of sugary drink cessation.

## 2.3. Data Analysis

Data were assessed for normality using a Shapiro–Wilk test, and non-parametric tests were used for measures with data that were not normally distributed. Descriptive statistics, including means and standard deviations, and counts and frequencies were used to summarize the demographic data, as appropriate. The frequency of reported intake for each beverage category was multiplied by the quantity reported to estimate beverage intake in ounces per day. An internal consistency analysis, using Cronbach’s alpha, was performed on the 37-item daily feelings questionnaire, using Stata/IC (Version 16.1).

Means and standard deviations for each item on the daily feelings questionnaire were calculated at baseline and on each day of sugary drink cessation, and a three-day cessation average was computed. An overall score was computed by summing the response for each item, with higher scores indicative of more aversive responses. Mean values for each item and overall scores reported during sugary drink cessation (each day and the three-day average) were compared to baseline values, using a non-parametric pairwise test (Dunn’s test), with post hoc false-discovery rate adjustments to account for multiple comparisons. Given that some participants were siblings, we also conducted a sensitivity analysis with a linear mixed model. The model had two random intercepts to account for repeated measures within individuals and clustering of siblings within families, and the fixed effect was the day of the measurement. With the exception of the internal consistency analysis and the linear mixed model, RStudio was used for all analyses. Statistical significance was set at *p* < 0.05.

Qualitative interview transcripts were transcribed verbatim and subsequently coded by one member of the research team (C.R.) using NVivo™ (version 12; QSR International, Inc.; Burlington, MA, USA). A codebook was generated based on participants’ own words. Codes were modified and revised iteratively throughout the coding process, and new codes were added as they emerged. Codes were then organized into themes and subthemes, and representative quotes from participants and their parents were selected.

## 3. Results

### 3.1. Participant Characteristics

A total of 151 participants were enrolled in the study. One participant was determined to be ineligible (based on age) during the baseline visit and was subsequently excluded. The demographic characteristics of the 150 eligible participants who completed the study are summarized in Table 1. The sample comprised an equal number of female and male participants (75 females and 75 males), and children’s ages ranged from 8 to 14 years (mean= 11.4 years). The sample was racially and ethnically diverse: nearly half of the participants self-identified as being non-White or having more than one race, and 20% reported Hispanic ethnicity.

### 3.2. Habitual Beverage Intake

Participants’ usual beverage intake, as reported at baseline, is summarized in Table 2. Reported consumption of sugary drinks was high, with just under two-thirds (59.4%) of the participants reporting consumption greater than two servings per day. Seventy-seven percent (*n* = 114) of participants reported some caffeinated sugary drink consumption, and approximately a quarter (*n* = 35) of the full sample reported consumption of caffeinated sugary drinks daily. Participants reported consuming an average of five different types of sugary drinks (e.g., caffeinated soda, caffeine-free soda, juice drinks, 100% fruit juice, and sports drinks) over the course of a typical week.

While most children (*n* = 110) reported drinking water at least once daily, about one-fifth (*n* = 32) consumed less than one 8-ounce serving per day, and approximately 40 percent (*n* = 59) reported daily consumption of unsweetened milk or milk alternatives. Seventeen participants reported consumption of less than 12 ounces of sugary drinks per day during the baseline visit, despite their parent reporting that they consumed ≥12 ounces of sugary drinks per day during eligibility screening. A sensitivity analysis was conducted after removing these 17 individuals, but no meaningful changes were observed.

### 3.3. Daily Feelings Questionnaire

Based on the internal consistency analysis, three items on the 37-item daily feelings questionnaire were dropped due to a negative item-test correlation. The remaining 34 items had excellent internal consistency (α = 0.93). Ninety-four percent of participants (*n* = 141) self-reported adherence to sugary drink cessation on all three days of the intervention. Overall negative feelings scores were significantly lower during the three days of sugary drink cessation compared with baseline (Table 3). This was the case when comparing each day of sugary drink cessation to baseline, as well as for the three-day sugary drink cessation average (all *p*’s < 0.001). The sensitivity analyses to account for the presence of siblings in the sample yielded similar findings. A statistically significant inverse association was observed between overall feelings score and day of cessation, where each day was associated with a lower score compared to baseline (β = −3.2, standard error = 0.42, *p* < 0.001). Participants reported improvements in physical feelings, including being less achy, less tired, and yawning less frequently, as well as having less trouble sleeping. Improvements in mood, such as being less angry and less annoyed, were also reported during sugary drink cessation. Furthermore, participants reported improvements in interpersonal interactions during sugary drink cessation, including arguing with others less, getting in trouble less often, and getting mad at people less frequently. The magnitude of the reported differences ranged from −1.2 to 0.2 on the 5-point scale.

### 3.4. Qualitative Interview

The demographic characteristics of the subset of participants who participated in a qualitative interview (Table 1) were similar to those of the total study sample. Among qualitative interview participants, the sample consisted of approximately half females and half males ranging from 8 to 14 years old, with most participants self-identifying as either Black/African American (41.4%) or White/Caucasian (37.9%). Two overarching themes pertaining to children’s physical and emotional responses to sugary drink cessation emerged: favorable responses during sugary drink cessation (Table 4) and aversive responses during sugary drink cessation (Table 5).

As shown in Table 4, four subthemes reflecting favorable responses to sugary drink cessation were identified: (1) improved sleep, (2) increased energy, (3) improved mood, and (4) physical benefits. The children reported that they experienced improved sleep quality during the study, and the parents agreed that their children fell asleep and woke up more easily during the three days of sugary drink cessation. Children and their parents also reported that the participant had more energy in the absence of his or her usual sugary drink intake, as well as positive changes in mood. For example, children indicated feeling happier and less irritable during sugary drink cessation, and parents stated that their children were less combative, for example, in their interactions with siblings. Some children also reported that they felt calmer while participating in the study, which was corroborated by parents. Participants and their parents also perceived health benefits associated with study participation, such as clearer skin, cleaner teeth, and feeling better overall during the sugary drink cessation period.

In contrast to the favorable responses to sugary drink cessation described above, three subthemes emerged reflective of aversive feelings during sugary drink cessation (Table 5). Some children reported that they felt more tired because they were not able to have sugar to wake them up, and others reported missing caffeine. Several parents also indicated that avoiding caffeinated sugary drinks was particularly challenging for their children. Another subtheme was related to disturbances in the participant’s mood during sugary drink cessation. Some parents reported that their children complained more than usual and/or that their child was angrier and meaner during the three days of sugary drink cessation. Some children also reported being upset, feeling sad, and crying because they could not have sugary drinks. A few children reported experiencing unpleasant physical symptoms, such as headaches, stomachaches, and feeling sick. In addition, some children reported having cravings for sugary drinks, and these cravings were also described by their parents.

Additional minor themes identified pertained to factors that either challenged or facilitated adherence to sugary drink cessation. Most notably, participants described environmental factors, such as availability of sugary drinks in the home or seeing others drink sugary drinks, as challenges to adherence. Participants also described eating out as difficult, particularly at restaurants where meals come with a sugary drink. Children also described a preference for the taste of sugary drinks compared to water, which made adhering to sugary drink cessation more difficult. In contrast, however, many children reported that it was easy to replace sugary drinks with unsweetened beverages, and several children described enjoying the challenge of avoiding sugary drinks for three days. Participants reported that a key facilitator of intervention adherence was being held accountable by someone else in the household. For example, children found sugary drink cessation more manageable when their parent or sibling reminded them not to consume sugary drinks, and, in some cases, the parents reported they also refrained from sugary drink intake during the intervention to encourage adherence.

## 4. Discussion

Our results demonstrate that children who habitually consume sugary drinks reported overall improvements in their physical feelings and emotions during three days of sugary drink cessation. Children reported lower oppositionality during sugary drink cessation based on ratings for survey items pertaining to irritability (e.g., “I felt annoyed” and “I got mad at people”), argumentativeness (e.g., “I argued with others”), and less behavioral misconduct (e.g., “I got in trouble”). Children also reported improvements in physical feelings, including less restlessness (e.g., “I could not sit still”), being less tired, and having less trouble sleeping. In addition to feeling better overall, on average, children did not report increases in feelings of need (e.g., “I felt like I needed sugary drinks”) or desire for sugary drinks (e.g., “I really wanted sugary drinks”); rather, they reported asking for sugary drinks less during the three-day sugary drink cessation period.

Findings of “Stop the Pop” did not support our initial hypothesis that children would experience aversive physical and emotional responses during sugary drink cessation and are in contrast with the findings recently reported by Falbe et al. [13]. This may be explained by the older age and higher habitual sugary drink consumption of their participants (mean age of 15 years old and habitual consumption of ≥3 sugary drinks per day versus 11 years old and ≥1 sugary drink per day in “Stop the Pop”). The lack of uniform aversive responses to sugary drink cessation reported in our study may also be explained, in part, by inter-individual variability in responses to sugary drink cessation. Even among adults who are high consumers of caffeine (frequent consumption of which is known to cause physical dependence), aversive responses to caffeine cessation are reported only by some individuals; and the percentage of individuals who report experiencing withdrawal symptoms varies widely across studies [22].

Furthermore, despite the overall improvements in physical feelings and emotions reported by participants in “Stop the Pop,” favorable responses were not unanimous. Aversive responses, including feelings of tiredness, anger, sadness, and headaches, were reported by some participants in qualitative interviews conducted following the three days of sugary drink cessation. Future research should examine potential moderators, such as amount of habitual caffeine intake, that may contribute to inter-individual variability in responses to sugary drink cessation, and which may not have been captured when comparing mean differences.

Nonetheless, the overall favorable responses to three days of sugary drink cessation reported during “Stop the Pop” may have important implications for ongoing public health efforts to reduce sugary drink consumption among children. Improvements in physical feelings and emotions experienced by children during short-term sugary drink cessation may lay the groundwork for more sustained reductions in sugary drink intake. Favorable responses reported by children in response to sugary drink cessation could also be leveraged in sugary drink reduction interventions as a means of increasing children’s motivation to initiate sugary drink cessation. In fact, in our prior study among children of similar age who reported habitual daily consumption of caffeinated sugary drinks, some participants indicated that sugary drink consumption gave them stomachaches or made them feel jittery [12]. Associations of sugary drink intake with stomachaches and feelings of sickness among children have also been reported elsewhere [23], along with other negative consequences resulting from sugary drink intake, such as decreased energy and impaired athletic performance [23]. Highlighting these previously reported negative consequences of sugary drink intake and the present finding that children responded favorably to sugary drink cessation could increase the likelihood that children will adhere to recommendations to limit or avoid sugary drinks.

Despite the promise of our findings for future efforts to reduce children’s sugary drink consumption, the positive responses to sugary drink cessation observed during “Stop the Pop” may be partially attributable to social desirability bias and/or positive expectancy bias, which could not be controlled for given the inherent limitations of the single-arm design. Children are known to answer surveys in a socially desirable manner in the presence of their parents. This is particularly the case when questions are subjective in nature and/or in instances when the child perceives certain answers may result in unwanted consequences [24]. While the daily feelings questionnaire was completed by children with their parent present during the initial study visit, we did not assess whether the parent was present at the time of questionnaire completion during the three days of sugary drink cessation. Therefore, it is not possible to determine the extent to which the parent may have influenced the child’s responses on the questionnaire during sugary drink cessation. In fact, some children did not report any differences in physical feelings or emotions during the interview despite indicating feeling differently during cessation based on their daily feelings questionnaire responses.

Positive-outcome-expectancy bias may also, in part, explain improvements in physical feelings and emotions reported by children during sugary drink cessation. Outcome expectancy, defined as the perceived consequences of a behavior, posits that an individual’s decision to engage in a behavior depends on whether he or she appraise its consequences as beneficial or detrimental [25]. In the context of sugary drink cessation, children’s knowledge of the negative consequences of sugary drink intake, as supported by our previous work [16], may have led them to expect beneficial effects of avoiding sugary drinks. Similarly, in qualitative interviews previously conducted with parents whose children habitually consume sugary drinks, most expressed concern about their children’s sugary drink intake, and many claimed to voice this concern to their children and/or actively attempt to lower their child’s sugary drink consumption [12,23]. It is therefore plausible that children may have perceived sugary drinks as negatively affecting their behavior and thus may have had a preconceived expectation that they would feel better when not consuming sugary drinks.

While improvements in physical feelings and emotions reported during three days of sugary drink cessation may encourage more prolonged sugary drink avoidance among children, our study was subject to several limitations. Although caffeinated and caffeine-free sugary drinks were differentiated when assessing children’s habitual beverage intake, daily consumption of caffeinated sugary drinks was relatively low in the sample (*n* = 35 participants), and our study was not powered to examine differences in responses to sugary drink cessation based on caffeine intake. Given that many sugary drinks contain added sugar and caffeine, both of which are highly rewarding ingredients, future studies to examine the extent to which heterogeneity in children’s responses to sugary drink cessation may be related to differences in their habitual caffeine intake are needed. Furthermore, no information on children’s sleep duration or quality, which may have impacted their perceived energy and/or tiredness, was collected. Children’s anthropometry was also not assessed. Changes in overall dietary intake during the study were also not evaluated; therefore, the extent to which children may have increased consumption of added sugar from other sources during sugary drink cessation cannot be ascertained. However, during the qualitative interviews, some participants described compensating for reduced sugar from sugary drinks by increasing sugar intake from food when asked about whether they noticed/made any dietary changes during the intervention. Inclusion of more comprehensive dietary assessments both at baseline and during sugary drink cessation in future studies will allow for an examination of possible substitution effects. Next, a more prolonged baseline assessment period (several days instead of a single day) would also have more accurately captured children’s physical feelings and emotions prior to beginning sugary drink cessation. In the same vein, the short duration of this study precludes us from understanding longer-term effects of sugary drink cessation. Finally, while our single-arm design allowed for the assessment of children’s feelings before versus after sugary drink cessation, the lack of a control group is an important limitation and precludes determination of the extent to which the reported improvements may be explained by measurement reactivity [26]. Taken together, the present findings should be interpreted cautiously, as improvements following sugary drink cessation may be overestimated.

It is also important to mention that the COVID-19 pandemic may also have affected the study findings; “Stop the Pop” was conducted entirely virtually throughout the pandemic. In the qualitative interviews, most participants described the COVID-19 pandemic as having substantially changed their usual routines, which may have impacted children’s responses to sugary drink cessation. Interestingly, some participants reported the pandemic made cessation more challenging because sugary drinks were easily accessible at home, while others reported that being at home made adherence to sugary drink cessation easier due to spending less time with friends and/or eating at restaurants less frequently. Regardless, reported adherence to sugary drink cessation during “Stop the Pop” was extremely high (94%).

Along with the high reported adherence to sugary drink cessation, another strength was the study’s diverse sample of participants, who were recruited from throughout the continental US and represented 31 states, along with the District of Columbia. The study was also strengthened by the use of a mixed-method research design that combined quantitative survey responses with semi-structured qualitative interviews conducted in a subset of participants. Finally, the study’s attrition rate was 0%.

## 5. Conclusions

Taken together, the study findings demonstrate that children who habitually consume sugary drinks experience favorable consequences of short-term sugary drink cessation; these results may serve to lay the groundwork for adherence to more prolonged sugary drink cessation. The results of “Stop the Pop” also call attention to the marked heterogeneity in children’s responses to sugary drink cessation, especially considering the divergent findings reported in a previous study utilizing a similar design [13]. Future research is needed to more robustly examine heterogeneity in children’s physical and emotional responses to sugary drink cessation, including effects of potential moderators, such as caffeine. Additional studies, which should extend the length of the cessation period, are needed to inform the development of targeted intervention strategies to reduce children’s sugary drink intake and allow for the identification of subsets of children at high-risk of facing challenges with sugary drink reduction.

## Figures and Tables

**Table 1 nutrients-14-01328-t001:** Demographic characteristics of “Stop the Pop” participants ^1^.

Characteristic	All	Subset that Completed the Qualitative Interview
N	150	30
Age (years), mean (sugary drink)	11.4 (2.0)	12 (1.6)
Sex		
Female	75 (50.0)	14 (46.7)
Male	75 (50.0)	16 (53.3)
Race ^2^		
White or Caucasian	76 (53.1)	11 (37.9)
Black or African American	43 (30.1)	12 (41.4)
Mixed race	18 (12.6)	6 (20.7)
Asian	6 (4.2)	0
Hispanic ethnicity	20 (13.3)	2 (6.7)
Eligibility for free or reduced price lunch ^3^		
No	71 (47.3)	17 (56.7)
Yes	66 (44.0)	10 (33.3)
Don’t know	13 (8.7)	3 (10.0)

^1^ Values are n (%), unless otherwise indicated. ^2^ *n* = 7 participants did not report their race including *n* = 1 in the qualitative interview. ^3^ Eligibility for free or reduced price lunch refers to the income-based provision of free or reduced price food at school assisted by the US Department of Agriculture; it is used as a proxy for relatively lower family income as opposed to ineligible families [21].

**Table 2 nutrients-14-01328-t002:** Participants’ habitual beverage intake, as reported at baseline, *n* = 148 ^1^.

Beverage Intake	*n* (%)
Daily 12-ounce sugary drink servings ^2^	
<1	17 (11.5)
1–2	43 (29.1)
>2–4	52 (35.0)
>4–6	18 (12.2)
>6	18 (12.2)
Daily 8-ounce plain water servings	
Never	1 (0.7)
<1	31 (20.9)
1–2	47 (31.3)
>2–4	36 (24.3)
>4–6	27 (18.2)
>6	6 (4.1)
Daily consumer of milk (yes) ^3^	59 (39.9)

^1^ Two participants had missing frequencies and/or quantities for their beverage intakes and wereexcluded from the analyses of the beverage intake questionnaire. ^2^ Includes both caffeinated and caffeine-free sugary drinks. ^3^ Includes any unflavored and unsweetened milk, including milk alternatives such as almond, soy, and oat milk.

**Table 3 nutrients-14-01328-t003:** Physical and emotional feelings reported on the daily feelings questionnaire at baseline and during three days of sugary drink cessation; *n* = 149 ^1^.

	Baseline	Cessation Average	Cessation Day 1	Cessation Day 2	Cessation Day 3
Overall score ^2,3^	64.9 ± 19.9	55.6 ±19.4 ***	56.4 ± 22.0 ***	55.7 ± 21.5 ***	54.6 ± 18.9 ***
I felt achy or sore.	1.5 ± 0.8	1.3 ± 0.6 **	1.3 ± 0.7 **	1.3 ± 0.7 **	1.3 ± 0.6 **
I felt angry.	1.9 ± 1.0	1.7 ± 0.8 *	1.7 ± 0.9 *	1.7 ± 1.0 *	1.7 ± 1.0 *
I felt annoyed.	2.1 ± 1.2	1.8 ± 0.8 *	1.8 ± 1.1 **	1.8 ± 1.1 *	1.7 ± 0.9 **
I argued with others.	2.2 ± 1.1	1.7 ± 0.7 **	1.7 ± 0.9 ***	1.6 ± 0.9 ***	1.7 ± 0.8 ***
I asked for sugary drinks.	2.9 ± 1.3	1.7 ± 1.0 ***	1.7 ± 1.1 ***	1.8 ± 1.2 ***	1.8 ± 1.2 ***
I was in a bad mood.	1.8 ± 0.9	1.7 ± 0.8	1.7 ± 1.0	1.7 ± 1.0	1.7 ± 0.9
I did not want to talk to people.	1.8 ± 1.1	1.5 ± 0.8	1.5 ± 1.0	1.5 ± 0.9	1.5 ± 0.8
I did not feel like doing anything	2.1 ± 1.3	1.7 ± 0.9 **	1.7 ± 1.1 **	1.8 ± 1.1 *	1.7 ± 1.0 **
I felt tired.	2.5 ± 1.2	2.1 ± 0.9 *	2.1 ± 1.1 *	2.1 ± 1.1 *	2.1 ± 1.1 *
I got mad at people.	2.0 ± 1.1	1.7 ± 0.9 **	1.7 ± 1.0 **	1.7 ± 1.1 **	1.7 ± 0.9 **
I felt grumpy.	1.8 ± 0.9	1.7 ± 0.8	1.7 ± 0.9	1.7 ± 1.0	1.6 ± 0.9
I had headaches.	1.5 ± 0.9	1.4 ± 0.7	1.4 ± 0.9	1.4 ± 0.8	1.4 ± 0.8
I had stomachaches.	1.4 ± 0.8	1.2 ± 0.5	1.2 ± 0.5	1.2 ± 0.6	1.2 ± 0.6 *
I had a hard time paying attention.	2.0 ± 1.1	1.8 ± 0.9	1.9 ± 1.1	1.8 ± 1.0 *	1.7 ± 0.9 **
I could not sit still.	2.4 ± 1.4	1.7 ± 0.9 ***	1.8 ± 1.1 **	1.6 ± 1.0 ***	1.7 ± 1.1 ***
I did not want to listen to my parents or other adults who take care of me.	1.8 ± 1.1	1.5 ± 0.8	1.5 ± 1.0 *	1.5 ± 0.9 *	1.5 ± 0.9
I felt weak.	1.4 ± 0.8	1.3 ± 0.6	1.4 ± 0.8	1.3 ± 0.8	1.3 ± 0.7
I had trouble sleeping.	2.0 ± 1.2	1.7 ± 0.9 *	1.7 ± 1.0 *	1.6 ± 1.0 *	1.7 ± 1.0 *
I wanted to be alone.	1.9 ± 1.1	1.7 ± 0.9	1.7 ± 1.1	1.7 ± 1.0	1.6 ± 1.0
I was mean to people.	1.6 ± 0.9	1.4 ± 0.7	1.4 ± 0.8	1.4 ± 0.8	1.4 ± 0.8
I was sleepy.	2.4 ± 1.2	2.1 ± 0.9	2.2 ± 1.1	2.1 ± 1.1	2.0 ± 1.0 *
I felt like I needed sugary drinks.	2.0 ± 1.2	1.9 ± 1.1	1.9 ± 1.2	1.9 ± 1.2	1.9 ± 1.2
I felt nervous.	1.7 ± 0.9	1.4 ± 0.7 *	1.5 ± 0.9	1.4 ± 0.7 *	1.4 ± 0.7 *
I did not feel like myself.	1.3 ± 0.7	1.4 ± 0.8	1.4 ± 1.0	1.4 ± 0.9	1.4 ± 0.9
I had other people get me sugary drinks.	2.1 ± 1.2	1.1 ± 0.4 ***	1.2 ± 0.7 ***	1.1 ± 0.5 ***	1.1 ± 0.5 ***
I felt sad.	1.6 ± 0.9	1.4 ± 0.7	1.5 ± 0.9	1.4 ± 0.8	1.4 ± 0.7
I felt sick.	1.3 ± 0.6	1.3 ± 0.6	1.3 ± 0.8	1.3 ± 0.8	1.3 ± 0.7
I tried to sneak sugary drinks.	1.5 ± 1.1	1.3 ± 0.7 *	1.2 ± 0.8 **	1.3 ± 0.8 **	1.3 ± 0.9 *
I felt stressed out.	1.7 ± 1.0	1.6 ± 0.8	1.6 ± 0.9	1.6 ± 1.1	1.5 ± 0.9
I got in trouble.	1.8 ± 1.0	1.4 ± 0.7 **	1.4 ± 0.8 **	1.4 ± 0.7 **	1.5 ± 0.8 **
I really wanted sugary drinks.	2.6 ± 1.3	2.4 ± 1.2	2.4 ± 1.3	2.4 ± 1.3	2.3 ± 1.3
I really wanted sugary foods.	2.3 ± 1.3	2.1 ± 1.1	2.2 ± 1.2	2.2 ± 1.3	2.0 ± 1.2
I was upset about not getting sugary drinks.	1.5 ± 1.0	1.8 ± 1.1	1.7 ± 1.1	1.8 ± 1.2	1.8 ± 1.2
I yawned.	2.5 ± 1.2	2.0 ± 0.9 *	2.2 ± 1.2 *	2.1 ± 1.1 **	1.9 ± 0.9 **

^1^ All values are reported as mean ± standard deviation. ^2^ *n* = 149, and one participant had missing data for one day of sugary drink cessation and therefore was not included in the calculation of the overall score. ^3^ Possible overall scores on the daily feelings questionnaire range between 34 (minimum) and 170 (maximum), where higher scores indicate more aversive feelings. *** *p* < 0.001, ** *p* < 0.01, and * *p* < 0.05; Dunn’s test comparisons between baseline and each day.

**Table 4 nutrients-14-01328-t004:** Favorable responses to sugary drink cessation reported during qualitative interviews with participants and their parents; *n* = 30 participants.

Theme Subtheme	Representative Quotations Child	Parent
Improved sleep	“I slept better. That was different. I slept good. Honestly, I feel like not having any sugary drinks made me like sleep better because most of the time I have a hard time sleeping, but I slept really good”.	“Sleep-wise when I tell them it’s time to go to sleep, they go right to sleep. They wake up more easily in the morning”.
Increased energy	“I’ve been really energetic the last few days”. “And I found it pretty easy to wake up with just like, water and my normal breakfast”.	“I think he had a little more energy, he’s usually pretty lethargic when he’s doing his home studies, but he seemed a little more alert”.
Improved mood		
Feeling happier	“I felt really, really, really happy”.	“They were not as cranky, grumpy-ish. Just I mean they seemed happier”.
Less irritable	“I wasn’t as irritated as I usually am. I wasn’t as aggressive as I usually am”.	“She’s been pleasant enough to be around. So, I don’t think it’s been a negative–I don’t think it’s been a negative thing at all”.
Less arguing	None	“There wasn’t all that sibling arguing and bickering that I would hear. They just seemed to get along a whole lot better”.
Calmer	“When I drank water, I was calmer and I wasn’t that hype”.	“She was a little calmer–not much calmer, but a little”.
Physical benefits		
Weight loss	“I’ve been losing weight”.	None
Clearer skin	“My skin was getting a little clearer”.	“I did notice her skin was like much clearer over these three days of just not having any of those [sugary drinks]”.
Cleaner teeth	“Our teeth are much cleaner”.	
Feeling better	“I felt a little bit better. I was kind of glad to change my diet up a little bit and drink a lot more water”.	“As the days went on, he started feeling better”.

**Table 5 nutrients-14-01328-t005:** Aversive responses to sugary drink cessation reported during qualitative interviews with participants and their parents; *n* = 30 participants.

Theme Subtheme	Representative Quotations Child	Parent
Less energy		
Feeling tired	“I felt tired because I didn’t have the sugar to like, give me energy and keep me awake in the morning”. “Um, just like really tired, lazy, I could like barely open my eyes”. “Yes, because Sprite wakes me up and water doesn’t”.	“She did mention that she felt a little tired in morning”.
Lower energy	“I felt kind of weak”. “It just felt–I just felt a little bit more–less energized. I’m not energized, but a little less–wanting to maybe, go do things, like not as active maybe”.	“She seems to be really kind of dragging to me”.
Wanting caffeine	“[Not having] the caffeine in the morning was probably the worst part”. “I felt tired because I didn’t have caffeine”.	“I don’t think she’s tempted to cheat. She’s just ready for something with caffeine”.
Worse mood		
Being annoying	“I was complaining a lot”.	“She’s getting on my nerves”. “He was kind of annoying to me”.
Being angry or mean	“I was a little bit grumpier”. “I was mad”.	“She’s been cranky since they haven’t been able to go to school and without sugar, oh my god, she’s even extra cranky, I would say. Like, saying it’s not fair”. “I got some mood change. He wasn’t as friendly. He was kind of, I would say, kind of mean”.
Feeling sad	“I bought a drink and I’m a little sad that I couldn’t have it”.	“She’s crying, well fake crying”.
Aversive physical changes		
Headaches	“I had slight headaches and that was about it”.	“You said you had a headache and so I thought maybe that could have been linked to it”.
Feeling sick	“I felt a little sick”. “I had stomachaches sometimes”.	None
Cravings	“When I was craving a drink, or like, a sugary drink that I was like thinking about drinks that were like similar that I could drink that didn’t have any sugar”.	“I know she’s been craving sugar”.

## Data Availability

The datasets generated and analyzed during the current study are available from the corresponding author upon reasonable request.

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
