# Peer review of "Stop the Pop: A Mixed-Methods Study Examining Children’s Physical and Emotional Responses during Three Days of Sugary Drink Cessation"

_nutrients, 2022, doi:10.3390/nu14071328_

Round 1

Reviewer 1 Report

The authors investigated the effect of SSB cessation on physical and emotional responses in American children using a questionnaire and interview. Although the topic is potentially important and interesting, there are many serious concerns that should be sufficiently addressed to improve the quality of the manuscript.

Major

1) The most serious concern of this study is single-arm design without control. Therefore, as the authors mentioned, they could not estimate the actual effect of the SSB cessation. Because participating children and parents were considered to expect the benefit of session, at the same time, give biased response owing to social desirability, the control group is essential for this study.

Minor

2) Abstract

This section should be a total of 200 words maximum, according to the Instructions for Authors of this journal.

3) The authors should not use SD as an abbreviation for sugary drink. It seems to refer to standard deviation. Please revise similarly elsewhere.

Method

4) Participant recruitment and eligibility

Please state that the recruitment was conducted to include an equal number of boys/girls (and other characteristics, if any condition was set). The authors should clarify when they ceased recruiting participants (e.g. when a sufficient number of participants were recruited), but the authors should recruit more participants, considering the estimated sample size and drop-out rate. Additionally, they should include only participants who completed questionnaires (BEVQ-15 and 3 day-daily feeling questionnaires) and 3-day cessation to the final analysis because of the aim of this study. Therefore, the statement in line 382 is incorrect and should be deleted. A flowchart explaining the participant selection procedure may help readers' understanding.

5) Measures

Please explain the validity of BEVQ-15, CWSQ, and ProWSC.

6) Qualitative Interview

Please clearly state how the authors defined saturation of data (line 138) and provide the rationale for the definition.

7) Data analysis

The authors should show the results derived from the mixed model as the main results.

Results

8) Lines 183-184

...nearly 60% of the participants?

9) Tables

The title of tables should be self-explanatory, i.e. they should include information on the number and characteristics of study participants. Statistical analysis conducted should be explained in each footnote. Table 5 is cut off in the middle.

Discussions

10) Lines 301-303

This statement should be interpreted with caution owing to the present study design.

11) Lines 316-367

The authors should state that these descriptions are the limitations of the present study.

Reviewer 2 Report

General Comment:

The manuscript is relevant to the current need to support policy formulation specifically on sugar reduction program of developed and developing countries. Scientific basis of policy formulation is very important, hence soundness of the research design and interpretation of result is very important.

On the introduction:

A brief review of literature on the subject was presented by the authors to illustrate the current state-of-knowledge on  the subject. The authors presented the research gap in particular what will be the outcome when children stop consuming sugary drink (SD) in particular on emotional and physical responses..

On the methodology:

The design of the study is a mixed design - qualitative and quantitative.  The research team implemented the data collection online/remotely.  However, the overall design of the project may not be enough to gather relevant and significant changes on the responses in the questionnaire.  Three days SD cessation may not reflect true change in behavior nor habit of the respondents. It is too short to examine how the subject behave during the SD cessation.  Studies showed that it will take 6 months to 1 year to see the real significant change in behavior of children.

If changes were noted during the period of intervention it may be due to some other factors like....conditions at home when data were collected, absence of the SD at home, restrictions in the movement of the subject, since most of them were at home due to COVUD 19, presence of guardians or parents, etc.

I suggestion if 3 days will be considered in the study, a strong justification on the 3 days intervention must be included in the manuscript.  Some ethical issues maybe behind the reason for shorter intervention period.

For appropriate guidance of the reader, it is suggested that the authors describe how the counseling was done during the intervention. Are they done by health professionals like nutritionist and dieticians?

Tables 1 and 2 can be further improved by segregating some data for clarity of the information/results.  It is difficult to comprehend.

What do you mean by "Eligibility for free or reduced-price lunch" ?

Table 3 not clear.  What do you mean by the numbers in the table?  e,g, overall score at baseline, cessation average etc.,,,,   64.9 ± 19.9 55.6 ±19.4*** 56.4 ± 22.0*** 55.7 ± 21.5 54.6 ± 18.9 etc .  I suggest to authors to provide a legend to understand the meaning of the numbers

Table 4 and 5.  Please provide clearer presentation of the tables to appreciate results

I noted that their are more limitations in the study.  These limitations could have been addressed by proper experimental design.

The paper has missing section - Conclusion. - What are the significant findings?

The authors may consider in the future study to include biochemical indicators responsible for the children's active/in active behavior, e.g. hormones etc.

Round 2

Reviewer 1 Report

Although the authors well-revised the manuscript according to my comments, some concerns remain that should be addressed sufficiently.

1) Participant recruitment and eligibility

The authors have stated that they ended participant recruitment once 150 participants were enrolled, but they recruited 151 participants (Line 189). Please explain.

I have recommended the authors include only participants who completed both questionnaires and 3-day cessation to the final analysis according to the aim of this study. Thus, participants included should be less than 141. Additionally, how about a flowchart explaining the participant selection procedure? The authors do not have to accept all of my comments, but they clearly should justify the reason the revision was not made.

2) Measures

For BEVQ-15, CWSQ, and ProWSC, please provide information on the validity (e.g. correlation coefficients or Cronbach α) to help readers' understanding.

Reviewer 2 Report

Dear Authors,

Thank you for considering my comments and suggestions.  The paper improved significantly.  Just minor suggestions for the improvement of the manuscript.  In Tables 4 and 5.  You may consider adding two columns, one for the response of the child and the other column for the response of the parents. This will help the reader understand your results.

Thank you
